# Predictors of Mortality in Hemodialyzed Patients after SARS-CoV-2 Infection

**DOI:** 10.3390/jcm11020285

**Published:** 2022-01-06

**Authors:** Leszek Tylicki, Ewelina Puchalska-Reglińska, Piotr Tylicki, Aleksander Och, Karolina Polewska, Bogdan Biedunkiewicz, Aleksandra Parczewska, Krzysztof Szabat, Jacek Wolf, Alicja Dębska-Ślizień

**Affiliations:** 1Department of Nephrology Transplantology and Internal Medicine, Medical University of Gdańsk, 80-210 Gdańsk, Poland; ptylicki@gumed.edu.pl (P.T.); aleksanderoch@gumed.edu.pl (A.O.); kpolewska@gumed.edu.pl (K.P.); bogdan.biedunkiewicz@gumed.edu.pl (B.B.); adeb@gumed.edu.pl (A.D.-Ś.); 27th Naval Hospital in Gdańsk, 80-305 Gdańsk, Poland; e.puchalska@7szmw.pl (E.P.-R.); puchola@gmail.com (A.P.); k.szabat@7szmw.pl (K.S.); 3Department of Hypertension and Diabetology, Medical University of Gdańsk, 80-210 Gdańsk, Poland; jacek.wolf@gumed.edu.pl

**Keywords:** COVID-19, SARS-CoV-2, hemodialysis, chronic kidney disease, mortality

## Abstract

Introduction: The determinants of COVID-19 mortality are well-characterized in the general population. Less numerous and inconsistent data are among the maintenance hemodialysis (HD) patients, who are the population most at risk of an unfavorable prognosis. Methods: In this retrospective cohort study we included all adult HD patients from the Pomeranian Voivodeship, Poland, with laboratory-confirmed SARS-CoV-2 infection hospitalized between 6 October 2020 and 28 February 2021, both those who survived, and also those who died. Demographic, clinical, treatment, and laboratory data on admission, were extracted from the electronic medical records of the dedicated hospital and patients’ dialysis unit, and compared between survivors and non-survivors. We used univariable and multivariable logistic regression methods to explore the risk factors associated with 3-month all-cause mortality. Results: The 133 patients (53.38% males) aged 73.0 (67–79) years, with a median duration of hemodialysis of 42.0 (17–86) months, were included in this study. At diagnosis, the majority were considered to have a mild course (34 of 133 patients were asymptomatic, another 63 subjects presented mild symptoms), while 36 (27.07%) patients had low blood oxygen saturation and required oxygen supplementation. Three-month mortality was 39.08% including an in-hospital case fatality rate of 33.08%. Multivariable logistic regression showed that the frailty clinical index of 4 or greater (OR 8.36, 95%CI 1.81–38.6; *p* < 0.01), D-Dimer of 1500 ng/mL or greater (6.00, 1.94–18.53; *p* < 0.01), and CRP of >118 mg/L at admission (3.77 1.09–13.01; *p* = 0.04) were found to be predictive of mortality. Conclusion: Very high 3-month all-cause mortality in hospitalized HD patients was determined mainly by frailty. High CRP and D-dimer levels upon admission further confer mortality risk.

## 1. Introduction

A population with a relatively high incidence of COVID-19, and associated high in-hospital mortality, is that of chronically hemodialyzed (HD) patients [1]. They are often elderly, have a high rate of comorbidities, and lowered immunity, which puts them at risk of both SARS-CoV-2 infection and a severe course of COVID-19. In the latest European Renal Association COVID-19 Database (ERACODA) report, their 28-day probability of death is 25% for all HD patients, and 33.5% for individuals who were admitted into hospitals [2]. In one of our previous surveys we found the unusually high in-hospital mortality of COVID-19 HD patients from the north of Poland, with a fatality rate up to 43.81% in subjects over 74 years old [3]. In another survey, we found that three months after eradication only 6.3% of HD patients were completely free of any COVID-19-related symptoms. Despite the significant improvement observed between months 3 and 6 of follow-up; a lot of cured HD patients still experience persistent symptoms [4]. COVID-19 may affect the cardiovascular system, and post-recovery may lead to myocarditis, arrhythmias, and thromboembolic events in individuals from the general population with, and without, preexisting cardiovascular disease. Cases of sudden cardiac arrest in patients after recovery from COVID-19 have been described as well [5]. It can’t be ruled out, that a similar situation occurs in HD patients with a high burden of cardiovascular disease.

In the absence of effective COVID-19 treatment, vaccination is the only chance to improve the extremely poor prognosis in HD patients [6]. On the other hand, the threat does not completely disappear even when completely vaccinated. Recent studies show an increasing frequency of breakthrough infections in vaccinated HD patients, which may be associated with the emergence of new threatening variants and the waning of post-vaccination immunity over time [7]. In the present study, therefore, we return to concentrating on the issue of the fatality rate due to COVID-19 among maintenance HD patients, and we focus on identifying factors determining their mortality up to 3 months after discharge from the hospital. This issue has not been addressed in previous studies. The results of the study can help identify the most at-risk HD patients, who require the greatest attention in the management period during hospitalization and after discharge.

## 2. Materials and Methods

### 2.1. Setting

The Pomeranian Voivodeship is located in the northern part of Poland. The population of the Pomeranian Voivodeship was 2,343,928 inhabitants, including 1,881,844 adults on 31 December 2019. A total of 1319 patients were on chronic dialysis at this time, with about 95.3% receiving hemodialysis, and 4.7% receiving peritoneal dialysis. Chronically HD adult patients from the Pomeranian Voivodeship were treated in 15 dialysis units, of which 4 are public and 11 private [3]. Through a decision of the health authorities, all HD patients in the Pomeranian Voivodeship with SARS-CoV-2 infection before the vaccination era were obligatorily hemodialyzed in a dedicated unit at the 7th Naval Hospital. Depending on the severity of the COVID-19, the patients were hospitalized at the 7th Naval Hospital in Gdansk or in isolation close to the hospital’s dialysis unit. They were discharged from the isolation/hospital after two negative swabs.

### 2.2. Design

This was a retrospective cohort study of all adult maintenance hemodialysis patients from the Pomeranian Voivodeship, Poland, with a diagnosis of SARS-CoV-2 infection over a time period of five months from 6 October 2020 to 28 February 2021, aimed to find the predictors of all-cause mortality recorded up to 3 months after virus eradication. Importantly, the study was conducted before widespread vaccination against COVID-19 was introduced. The study received ethical approval in the Medical University of Gdansk (NKBBN/2014/2021).

### 2.3. Study Population

The study included patients tested positive for SARS-CoV-2 by RT-PCR who were then dialyzed in the 7th Naval Hospital in Gdansk.

### 2.4. Data Collections and Procedures

The records of patients were collected from the hospital’s electronic database by trained medical staff. The final data was verified by the major investigators. The collected data included demographic information, comorbidities, medications, history of dialysis treatment, symptoms on admission, initial clinical findings at physical examination, initial laboratory assessment, and COVID-19 severity at admission. The Charlson comorbidity index (CCI) was calculated by summing the assigned weights of all comorbid conditions presented by the patients, according to the original formula of Charlson et al., on admission [8]. The age-adjusted CCI (ACCI) incorporates age as a correction variable of the final score by adding 1 point for every decade over 40 years old [8,9]. The frailty index was calculated on a scale of 1–9, according to the Clinical Frailty Scale (CFS), on admission. The CFS applies functional descriptors and pictographs. An index of 1 represents very fit, and 9 represents terminally ill [10]. Disease severity was defined according to the clinical presentation of the individual at hospital/isolation admission, and was classified into four categories: (a) asymptomatic patients, (b) symptomatic patients without dyspnea, (c) patients with oxygen saturation below 93% requiring oxygen support, and (d) patients who were hypoxic at admission and required the intensive care unit. Follow-up surveys were completed by telephone interview with the staff of the home dialysis centers at the end of the third month after the patient’s discharge.

### 2.5. Outcome

The main aim of the study was to find predictors of 3-month all-cause mortality after SARS-CoV-2 infection, including hospital deaths and deaths after discharge.

### 2.6. Statistical Analyses

Quantitative variables were presented as medians (IQR), and qualitative variables were presented by frequencies and percentages. There was no imputation for missing data. The Fisher’s exact test, Chi Square or Mann-Whitney U tests were used to analyze differences between groups where appropriate. Risk factors associated with death and their odds ratio (OR) were calculated by a univariable logistic regression model. Multivariable logistic regression model was applied to identify independent predictors of mortality. To avoid overfitting of the regression model because of the small number of endpoint events (53), we arbitrarily defined variables to finally enter the multivariable analysis [11]. For practical reasons, clinical data which were initially presented as continuous variables and were confirmed in univariable analyses to increase OR for 3-month mortality—prior to multivariable modeling were dichotomized (this holds true to CFS, age, SpO2, D-Dimer, and CRP). The cut-off for each dichotomization was identified with ROC analyses, and Youden indices. We excluded from the logistic models variables if the data were not available >50% patients (some laboratory data) or if their nature was highly subjective (dyspnea). Finally, age, blood group 0, CFS, treatment with active vitamin D, CRP and D-dimer levels at admission were chosen. We assessed the model’s goodness-of-fit using the Hosmer-Lemeshow goodness-of-fit statistic.

## 3. Results

### 3.1. Patients Demographic and Clinical Characteristic

Between 6 October 2020 and 28 February 2021, a total of 154 HD patients with SARS-CoV-2 infection were referred to our institution. The cohort includes 133 of them from whom complete data were available. Table 1 presents a detailed breakdown of baseline characteristics, including demographics, comorbidities, and home drug treatments. The median age of the cohort was 73.0 (67–79) years, and 53.38% were males. Hypertension was the most common comorbidity, present in 128 of 133 patients, followed by diabetes in 73 individuals. The median duration of hemodialysis was 42.0 (17–86) months, and the most common vascular access was a dialysis catheter.

### 3.2. Presentation at Admission and Outcome

The median time from onset of illness to hospitalization was 1 (0–3) days. At admission, the majority were considered to have a mild course (34 of 133 patients were asymptomatic, another 63 subjects presented mild symptoms), while 36 (27.07%) patients had low blood oxygen saturation and were required to have oxygen supplementation. The median duration of hospitalization was 14.5 (9–20) days. Compared to the survivors, non-survivors were older and had a higher CCI, and CFS. They suffered from chronic pulmonary diseases more often, but there was no difference in the incidence of diabetes and cardiovascular diseases between these groups. Survivors significantly more often presented with blood group 0 and received active vitamin D (Table 2). Compared to survivors, non survivors had a worse clinical presentation at admission, reported dyspnea more frequently, had significantly higher levels of CRP and D-dimer, and lower blood oxygen saturation in initial lab values (Table 3).

### 3.3. Predictors of Mortality

Three-month mortality was 39.08% including an in-hospital case fatality rate of 33.08%. The main reason for in-hospital death was respiratory failure, followed by cardiovascular events. The median time from admission to in-hospital death was 8 (4–15) days. The cause of death after discharge in all nine non-survivors was sudden cardiac death.

The summary of univariate logistic regression is presented in the Figure 1. Frailty (CFS) appeared to be strongest sole predictor of 3-month mortality (OR = 11.7 95%CI 3.4–40.7; *p* < 0.001) followed by advanced age (OR = 8.0 95%CI 2.3–28.3; *p* = 0.001), whereas blood type 0 and chronic D-vitamin supplementation were associated with favorable prognosis (OR = 0.45, 95%CI 0.20–0.998; *p* = 0.49, and OR = 0.45 95%CI = 0.22–0.93; *p* = 0.03, respectively).

Owing to evident linear relationships between CCI and CFS (r = 0.323, *p* < 0.001) and CCI and age (r = 0.503, *p* < 0.001), CCI was excluded from further multivariavble analyses. Based on the results of the univariable logistic regreassion and arbitrary expert decision, a multivariable predictive model of a 3 month mortality was proposed. The following variables were included: age, CFS, chronic vitamin D supplementation, blood type, CRP and D-dimer. For practical reasons, all continuous variables were dichotomized as described in the method section. Appendix A) and Figure 2 summarize multivariable logistic regression (AUC = 0.84 95%CI 0.75–0.91; *p* < 0.0001).

Exploratory univariable analysis of factors associated with late death during 3 months after discharge was performed as well. As summarized in Appendix A) the CFS (*p* = 0.003), COVID-19 severity on admission (*p* = 0.03) and level of D-dimer (*p* = 0.01) were positively related to the mortality.

## 4. Discussion

Mortality from COVID-19 and its determinants is being carefully studied in the general population. Several studies have indicated that advanced age is the most important risk factor for mortality [12,13]. Across all age groups, male sex was found to be associated with an increased risk of mortality as well [13,14]. A severe course of COVID-19 is more likely to occur in patients with preexisting comorbidities, among which the most significant mortality predictor is chronic kidney disease [12,13,15,16]. Studies in the HD population on these important issues are sparse and show inconsistent results.

In the present study, including the cohort of all HD patients from dialysis units in Northern Poland, we have shown a 3-month mortality rate of 39.08% including an in-hospital case fatality rate of 33.08%. The mortality in previous studies in HD patients ranges greatly from 16.2% (Turkey) [17], through 20% (multinational survey) [18], 33.5% (ERACODA) [2], 39.2% (China) [19], to as high as 41% (Italy) [20]. This large discrepancy may be explained by the different ages of patients, different selection of patients (asymptomatic, hospitalized) and different observation periods between studies.

In the present study we identified frailty as the strongest risk factor for mortality in HD patients. In multivariable analysis the association between frailty and mortality was even stronger than the association between age and mortality. Our results in this regard are consistent with findings from the European Renal Association COVID-19 Database (ERACODA), specifically designed to prospectively collect detailed data on kidney transplant and dialysis patients with COVID-19 [2]. Frailty as a predictor of mortality is widely used in a variety of populations: older adults and nursing home residents [21], critically ill patients [22], and oncology patients [23]. Several reasons may account for this. First, patients with frailty suffer from a more vulnerable condition characterized by various observable deficits, such as a reduced physiologic reserve, chronic undernutrition and cognitive impairment, increasing the likelihood of an adverse outcome when patients are exposed to major negative stressors, including COVID-19. Second, frailty involves the process of complex chronic inflammation that exacerbates the risk of mortality when patients contract COVID-19. Finally, frail older people are often unable to endure intensive medical care, resulting in a greater likelihood of death during treatment [21].

In our study, advanced age has been identified as an independent risk factor for mortality yet the relationship was attenuated in multivariate model. This is a commonly reported relationship found both in the general population [24,25], and in all studies to date in maintenance HD patients [2,17,18,19,26]. Several reasons might contribute to this; including age-related physiological changes, impaired immune function, and pre-existing illnesses [24]. Male gender, and comorbidities such as diabetes were not associated with a high risk of death, which was inconsistent with previous studies in the general population [24,25]. Data among dialysis patients is inconclusive on this point. For instance, Alberici et al. reported that many HD patients had certain comorbid conditions such as cardiovascular disease, hypertension, diabetes, and lung disease, which were related to worse outcomes in patients with COVID-19 [27]. On the contrary, hypertension, diabetes mellitus, coronary artery disease, heart failure and chronic lung disease did not emerge as independent risk factors in data from the ERACODA database [2]. The analysis of our results leads us to believe that the impact of comorbidity on mortality may be very important. We found univariable associations of mortality with higher CCI, but it could not be included in the regression model due to unacceptable collinearity with frailty score and age.

Our findings confirm that initial laboratory assessment is very important for the risk stratification of COVID-19 patients and those demonstrating markers of inflammation, or coagulopathy are at increased risk of a poor outcome. As in the general population, CRP and D-dimer levels were associated with a significantly higher risk of death amongst COVID-19 patients. The inflammatory biomarker, CRP, was higher in deceased patients than in survivors, providing evidence for the presence of a cytokine storm that can contribute to the fatal outcome of many COVID-19 patients. It may also apply to the coagulation dysfunction common in patients with COVID-19, that may be especially due to the inflammatory responses induced by SARS-CoV-2 [28]. The association of the severity of inflammation and coagulopathy with mortality is the first such observation in HD patients. Previous studies in HD patients neither analyzed nor confirmed this relationship.

In univariable but not multivariable analyses we demonstrated for the first time in HD COVID-19 patients the protective effect of blood group O, and chronic treatment with active vitamin D. Some previous studies showed that type O blood may be associated with a lower risk for SARS-CoV-2 infection and severe COVID-19 illness or death in the general population while individuals with the A blood group are especially prone to developing the disease with unfavorable outcomes [29,30]. One of the reasons for such relationships may be that a non-O blood type is among the most important genetic risk factors for venous thromboembolism [31], and these conditions are relevant for the COVID-19 outcome. Low 25-hydroxy vitamin-D (25-OH D) levels are correlated with high IL-6 levels, and were found to be independent predictors of COVID-19 severity and mortality in the general population [32]. The evidence for the effectiveness of vitamin D supplementation for the prevention and treatment of COVID-19 is, however, very uncertain [33,34]. There was substantial clinical and methodological heterogeneity of the conducted studies, mainly because of different supplementation strategies, formulations, vitamin D status of participants, and reported outcomes [35]. Patients in our cohort did not have 25-OH-D levels determined, and, following the KDIGO guidelines, received 1-alpha-hydroxyvitamin D3 (Alfacalcidol), aimed to normalize calcium and phosphorus levels, and to maintain parathormone within two to nine times the upper normal limit [36].

Our study is the first to analyze 3-month COVID-19 mortality in hospitalized HD patients which is very important given that the disease does not end with the eradication of the virus. The persistent symptoms, and deterioration in the quality of life, may persist for up to 6 months after eradication and even for as long as 1 year [4,37]. The strength of our study is also that, unlike other studies, it represents the entire spectrum of the disease, from asymptomatic to severe cases in all HD patients from northern Poland, which additionally excludes any potential center effect. A limitation of this study is its observational design, which only permits the description of associations. Secondly, a relatively small sample for this type of research is another limitation of the study, which allows drawing conclusions of exploratory characteristics. Thirdly, our study does not include data on in-hospital patient management, which may have impacted the outcome. It cannot be also completely ruled out that the cause of deaths that occurred after discharge was independent of COVID-19. We believe that developing a predictive score system for mortality in this vulnerable population would be of great practical value. This is an extremely difficult challenge, however, given that the test group must be very large and the final model should also be subject to external validation by independent researchers before being made public. How difficult this task can be was shown in the review paper analyzing 50 such attempts in the general population. All models were considered to have a high risk of bias due to a combination of poor reporting and poor methodological conduct for participant selection, predictor description and statistical methods, and none were recommended for clinical use [38]. The solution may be the machine learning models using data from the national or continental databases.

## 5. Conclusions

In conclusion, 3-month mortality was very high in HD COVID-19 patients. Mortality was strongly associated with frailty, advanced age, markers of inflammation and coagulopathy. In addition, we found that among patients with blood group O and those who were treated with active vitamin D, there may be a lower risk of death from COVID-19. Understanding the clinical and laboratory predictors of mortality at diagnosis may help identify especially at-risk patients, and improve triaging and management throughout health systems.

## Figures and Tables

**Figure 1 jcm-11-00285-f001:**
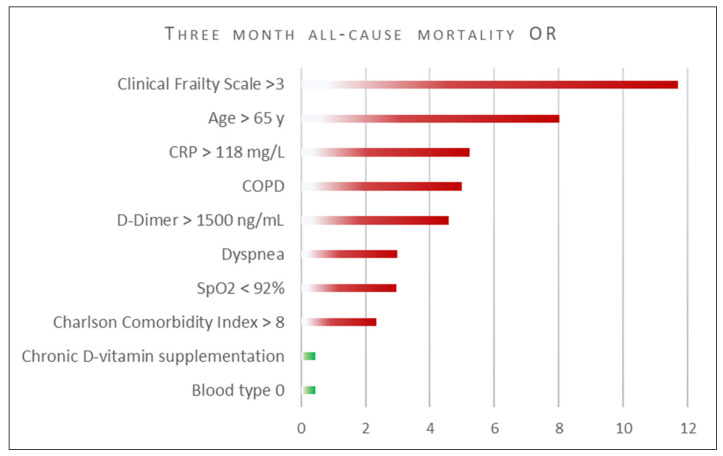
Clinical characteristics affecting 3-months mortality. Univariable logistic regression. Legend: COPD—chronic obstructive pulmonary disease; OR—odds ratio; SpO2—blood oxygen saturation (pulse oximetry finger probe). All lab test results obtained upon admission to the hospital.

**Figure 2 jcm-11-00285-f002:**
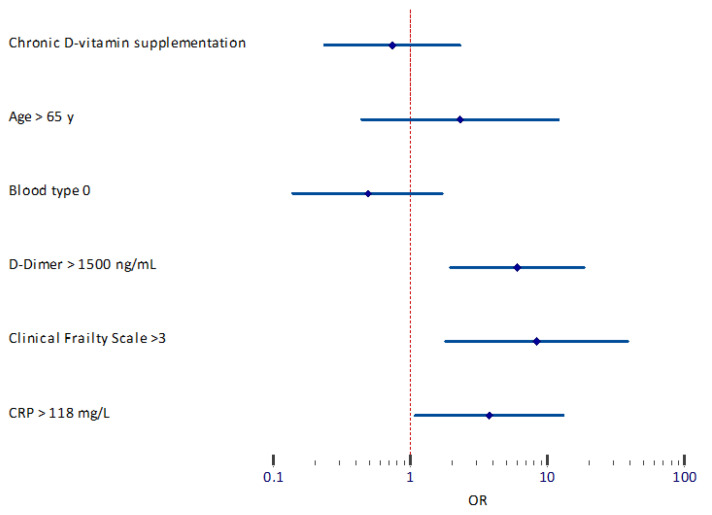
Predictors of 3-month mortality. Forest plot-multivariable logistic regression. Logarithmic scale.

**Table 1 jcm-11-00285-t001:** General characteristics of all study patients.

Variable	
*n*	133
Sex	
Male	71 (53.38)
Female	62 (46.62)
Age, years	73.0 (67–79)
Body mass index, kg/m^2^	26.0 (22.0–29.0)
Dialysis vintages, months	42.0 (17–86)
Dialysis dose per week, hours	12 (12–12)
Past kidney transplantation	11 (8.27)
Dialysis access	
AVF/AVG	46 (34.59)
Dialysis catheter	87 (65.41)
Comorbidities	
Diabetes	73 (54.88)
Hypertension	128 (96.24)
Chronic pulmonary disease	12 (9.02)
Ischemic heart disease	54 (40.60)
Congestive heart failure	53 (39.85)
Malignancy	15 (11.28)
Charlson comorbidity index (age adjusted)	8 (6–10)
Frailty index	4 (3–5)
COVID-19 severity on admission	
Asymptomatic	34 (25.56)
Mild symptomatic	63 (47.37)
Moderate requiring oxygen (<93%)	36 (27.07)
Oxygen saturation on admission (SpO2), %	94 (90–96)
Symptoms onset to hospital admission, days	1 (0–3)
Hospitalization duration, days	14.5 (9–20)
Died in the hospital	44 (33.08)
Respiratory failure	29
Cardiovascular reasons	9
Co-infections	4
Thromboembolic complications	1
Hemorrhagic complications	1
Time from admission to hospital death, days	8 (4–15)
Died during 3 months after discharge from hospital	9 (6.8)
Sudden cardiac death at home	9 (100)
Non survivors (died in the hospital or during 3 months)	53 (39.08)

Legend: AVF/AVG, arteriovenous fistula/graft.

**Table 2 jcm-11-00285-t002:** Demographic and baseline characteristics of HD patients with COVID-19 by patient outcome.

Variable	Survivors *n* = 80	Non-Survivors *n* = 53	*p*-Value
Sex			
Male	41 (51.25)	30 (56.6)	*p* > 0.1
Female	39 (48.75)	23 (43.4)	*p* > 0.1
Age, years	71.0 (61–77)	75.00 (70–81)	*p* < 0.001
Blood group:			
A	31(38.0)	24/50 (48.0)	*p* > 0.1
B	13 (16.25)	10/50 (20.0)	*p* > 0.1
AB	5 (6.25)	5/50 (10.0)	*p* > 0.1
0	31 (38.75)	11/50 (22.0)	*p* = 0.047
Body mass index, kg/m^2^	26.0 (22.0–29.0)	25.0 (22.0–29.0)	*p* > 0.1
Dialysis vintages, months	37.5 (17–90)	47 (17–82)	*p* = 0.09
Dialysis dose per week, hours	12 (12–12)	12 (12–12)	*p* > 0.1
Past kidney transplantation	7 (8.75)	4 (7.55)	*p* > 0.1
Comorbidities:			
Diabetes	42 (52.5)	31 (53.0)	*p* > 0.1
Hypertension	77 (96.25)	51 (96.25)	*p* > 0.1
Chronic pulmonary disease	4 (5.0)	8 (15.09)	*p* = 0.046
Ischemic heart disease	29 (36.25)	25 (47.17)	*p* > 0.1
Congestive heart failure	29 (36.25)	24 (45.28)	*p* > 0.1
Malignancy	6 (7.5)	9 (16.98)	*p* = 0.091
Charlson comorbidity index	7 (6–9)	8 (7–10)	*p* = 0.023
Clinical Frailty Scale (CFS)	4 (3–5)	5 (4–6)	*p* < 0.001
Medications:			
ACE inhibitors	15 (18.75)	11 (20.75)	*p* > 0.1
ARBs	8 (10)	2 (3.77)	*p* > 0.1
Calcium channel blockers	25 (31.25)	17 (32.07)	*p* > 0.1
Beta-blockers	59 (73.75)	36 (67.9)	*p* > 0.1
Statins	37 (46.25)	22 (41.51)	*p* > 0.1
Oral anticoagulant	6 (7.5)	2 (3.77)	*p* > 0.1
LMWH between dialysis days	17 (21.25)	15 (28.3)	*p* > 0.1
Active oral vitamin D	57 (71.25)	28 (53)	*p* = 0.03
Epoetin beta IU per week	6000 (3500–6000)	6000 (4000–9000)	*p* > 0.1

Data is *n* (%) or median (IQR), unless otherwise specified; HD, hemodialyzed; ACE, angiotensin converting enzyme; ARB, angiotensin II receptor blocker; LMWH, low molecular weight heparin.

**Table 3 jcm-11-00285-t003:** Pattern of presenting symptoms, and laboratory findings on hospital admission in survivors and non-survivors.

Variable	Survivors *n* = 80	Non-Survivors	*p*-Value
		*n* = 53	
Symptoms:			
Dyspnea	23 (28.75)	29 (54.72)	*p* = 0.003
Fever >37.3 °C	32 (40.0)	21 (39.62)	*p* > 0.1
Fatigue	36 (45.0)	21 (39.62)	*p* >0.1
Chills	13 (16.25)	7 (13.21)	*p* > 0.1
Cough	25 (31.25)	14 (26.41)	*p* > 0.1
Decreased appetite	4 (5.0)	4 (7.55)	*p* > 0.1
Diarrhea	9 (11.25)	4 (7.55)	*p* > 0.1
Headache	2 (2.5)	3 (5.66)	*p* > 0.1
Myalgia	3 (3.75)	1 (1.88)	*p* > 0.1
Chest pain	3 (3.75)	1 (1.88)	*p* > 0.1
Insomnia	1 (1.25)	1 (1.88)	*p* > 0.1
Smell or taste disturbances	4 (5.0)	1 (1.88)	*p* > 0.1
No symptoms	24 (30.0)	10 (18.87)	*p* = 0.15
Symptoms onset to admission, days	1 (0–3)	1 (0–3)	*p* > 0.1
COVID-19 disease severity:			
Asymptomatic-Mild	68 (85)	31 (58.5)	*p* < 0.001
Moderate-Severe	12 (15)	22 (41.5)	*p* < 0.001
Findings at physical examination:			
Heart rate, beats per min	80 (73–90)	84 (76–89)	*p* > 0.1
Respiratory rate, breaths per min	16 (15–17)	15.5 (14–20)	*p* > 0.1
Oxygen saturation, %	95 (92–97)	93 (86–96)	*p* = 0.005
Systolic blood pressure, mmHg	130 (122–152)	132 (116–150)	*p* > 0.1
Diastolic blood pressure, mmHg	80 (70–90)	75 (67–85)	*p* > 0.1
Temperature °C	36.7 (36.4–37)	36.8 (36.5–37.45)	*p* > 0.1
Laboratory findings:			
White blood cell count, ×10^9^/L	5.12 (3.99–8.11)	6.47 (4.71–9.08)	*p* = 0.05
Lymphocyte count/mm^3^	0.92 (0.63–1.4)	0.68 (0.56–0.98)	*p* = 0.08
Hemoglobin, g/dL	11.1 (10.0–11.6)	10.7 (9.3–11.6)	*p* > 0.1
Platelet count, ×10^9^/L	195.5 (149–239)	171 (129–230)	*p* > 0.1
CRP, mg/L	34.5 (9.1–95.1)	100.1 (38.1–174.4)	*p* < 0.001
D-dimer, ng/mL	1106.5 (724.9–1491.1)	1695.66 (897.3–3793.2)	*p* = 0.004
Procalcitonin, ng/mL	0.35 (0.26–0.64)	0.99 (0.69–4.22)	*p* > 0.1
Ferritin, ng/mL	1306 (777.9–1902.0)	1262.5 (479.4–2145.0)	*p* > 0.1
ALAT U/L	20.5 (12–30)	16 (12–28)	*p* > 0.1
pO2, mmHg	71.7 (54.2–87.6)	63.2 (47.4–73.9)	*p* = 0.1
pCO2, mmHg	33.85 (28.6–36.9)	34.75 (30.3–38.55)	*p* > 0.1

## Data Availability

Detailed data are available on request from the corresponding author.

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
