# Peer review of "Predictors of Mortality in Hemodialyzed Patients after SARS-CoV-2 Infection"

_jcm, 2022, doi:10.3390/jcm11020285_

Round 1

Reviewer 1 Report

This is a well written manuscript where the authors focus on identifying risk factors of mortality in hemodialyzed patients 3 months after discharge from the hospital.  See my comments to the authors below.

Authors found clinical and laboratory predictors of mortality such as frailty, advanced age, ABO blood group, CRP and D-dimer levels that may help identify at-risk HD covid patients.  In this study, sudden cardiac death at home is the main cause of death during 3 months after discharge from hospital, so I recommend to the authors should analyze these clinical and laboratory predictors of mortality with the sudden cardiac death at home to find some predictors of mortality after discharge.   

Author Response

Following the recommendations of the Reviewer, an analysis of factors that may be associated with late death after COVID-19 (after discharge) was performed. As in the case of total mortality, these factors included the disease severity on admission, the frailty clinical scale and the level of D-dimer. However, this analysis can only be treated as exploratory, due to the very small number of cases (9) and the large disproportion in numbers between  study  and  control group. For this reason, large differences between groups, e.g. in CRP, lymphocytes, age, treatment with active vitamin D did not reach statistical significance.  Inference from these calculations is of limited value, therefore it was decided to only make a small mention in the manuscript results section (line=208)  as follows:

“Exploratory univariable analysis of factors associated with late death during 3 months after discharge was performed as well. As summarized in Table S1 (in supplementary materials) the FCS (p=0.003), COVID-19 severity on admission (p=0.03) and level of D-dimer (p=0.01) were positively related to the mortality”

and to include an additional Table S1 in the supplementary materials.

Reviewer 2 Report

Thank you for the review.

The authors are doing a study to investigate the determinants of COVID-19 mortality among the maintenance hemodialysis (HD) patients, who are the population most at risk of an unfavorable prognosis. In this retrospective cohort study they included all adult HD patients from the Pomeranian Voivodeship, Poland, with laboratory-confirmed SARS-CoV-2 infection hospitalized between both those who survived, and also those who died.

The study is also very relevant as Demographic, clinical, treatment, and laboratory data on admission, was extracted from the electronic medical records of the patients dialysis compared between survivors and non-survivors.  Multivariable logistic regression showed that the frailty clinical , age , D-dimer, and c-reactive protein (CRP)  levels on admission were found to be predictive of mortality, while chronic active vitamin D treatment and blood group 0 were negatively associated with mortality. This does give a very good idea of the predictors of mortality and does go with the current data available.

The authors have concluded that, a very high 3-month all case mortality in hospitalized HD patients was determined by frailty and 34 older age. Blood group 0 and vitamin D treatment exerted protective effect, while high CRP and D-dimer levels were independent predictors of mortality.

I have one suggestion, is it possible for the authors to devise a predictive scale based on the above information, as that can lead to not only quick identification of the high risk HD patients but also quantify the risk.

The discussion is well written , and referenced. The language used is appropriate .

Thank you

Author Response

A very valuable comment. Like the Reviewer, we believe that developing a predictive score system for mortality would be of great practical value. However, this is an extremely difficult challenge. First, the test group must be very large. It is very difficult to achieve in the population of dialysis patients especially in single-center studies and therefore, no such predictive scales have been developed in this population so far. The solution may be the analysis of national or continental databases. The obtained results should be also subject to external validation by independent researchers before being made public. Otherwise, the model may be biased and unreliable and could cause more harm than benefit in guiding clinical decisions. How difficult this task can be is shown by the results of the review paper in the BMJ by Wynants L. et al analyzing 50 such attempts in the general population. All models were considered to have a high risk of bias due to a combination of poor reporting and poor methodological conduct for participant selection, predictor description and statistical methods, and none were recommended for clinical use. Perhaps the best solution would be to use machine learning models to identify drivers of progression to more severe disease and for mortality. These are the reasons why we were unable to take on this challenge. Noting the need to develop such a scale, a relevant commentary on this matter was included in the discussion (line 299) as follows:

„We believe that developing a predictive score system for mortality in this vulnerable population would be of great practical value. This is an extremely difficult challenge, however, given that the test group must be very large and the final model should be also subject to external validation by independent researchers before being made public. How difficult this task can be was shown in the review paper analyzing 50 such attempts in the general population. All models were considered to have a high risk of bias due to a combination of poor reporting and poor methodological conduct for participant selection, predictor description and statistical methods, and none were recommended for clinical use [38]. The solution may be the machine learning models using data from the national or continental databases”.